# PRE-TRAINED TRANSFORMERS AS PLUG-IN DEFENDERS AGAINST ADVERSARIAL PERTURBATIONS

## ABSTRACT

With more and more deep neural networks being deployed as various daily services, their reliability is essential. It is frightening that deep neural networks are vulnerable and sensitive to adversarial attacks, the most common one of which for the services is evasion-based. Recent works usually strengthen the robustness by adversarial training or leveraging the knowledge of an amount of clean data. However, retraining and redeploying the model need a large computational budget, leading to heavy losses to the online service. In addition, when training, it is likely that only limited adversarial examples are available for the service provider, while much clean data may not be accessible. Based on the analysis on the defense for deployed models, we find that how to rapidly defend against a certain attack for a frozen original service model with limitations of few clean and adversarial examples, which is named as **RaPiD** (**Ra**pid **P**lug-**in D**efender), is really important. Motivated by the generalization and the universal computation ability of pre-trained transformer models, we come up with a new defender method, **CeTaD**, which stands for **C**onsidering Pr**e**-trained **T**ransformers **a**s **D**efenders. In particular, we evaluate the effectiveness and the transferability of **CeTaD** in the case of one-shot adversarial examples and explore the impact of different parts of **CeTaD** as well as training data conditions. **CeTaD** is flexible for different differentiable service models, and suitable for various types of attacks.

## 1 ANALYSIS: THE DEFENSE FOR DEPLOYED SERVICE MODELS

It is found that trained neural network models are so vulnerable that they could not predict labels correctly when limited perturbations are added into the input examples (Goodfellow et al. (2014); Akhtar & Mian (2018); Chakraborty et al. (2018)). Such a method is called an evasion-based adversarial attack. Facing this challenge, recent works (Xu et al. (2023); Wang et al. (2023); Shi et al. (2021); Wang et al. (2022); Nie et al. (2022)) pay attention to getting robust models by leveraging the knowledge of clean data or running adversarial training.

Table 1: Comparison of conditions of **RaPiD** and recent works on adversarial defense. We focus on the needs of generating extra data, tuning the target service models, applying adversarial training, using information from clean data and whether it is plug-in.

| Case | Data Generation | Tuning Service | Adversarial Training | Clean Data | Plug-in |
|------|:---:|:---:|:---:|:---:|:---:|
| Wang et al. (2023) | ✓ | ✓ | ✓ | ✓ | ✗ |
| Xu et al. (2023) | ✗ | ✓ | ✓ | ✓ | ✗ |
| Shi et al. (2021) | ✗ | ✓ | ✗ | ✓ | ✗ |
| Wang et al. (2022) | ✗ | ✗ | ✗ | ✓ | ✗ |
| Nie et al. (2022) | ✗ | ✗ | ✗ | ✓ | ✗ |
| Xie et al. (2017) | ✗ | ✗ | ✗ | ✗ | ✓ |
| Xie et al. (2019) | ✗ | ✗ | ✓ | ✓ | ✓ |
| Ours | ✗ | ✗ | ✓ | ✗ | ✓ |

Nowadays, deep neural networks are employed as fundamental services in various fields (Liu et al. (2017); Eloundou et al. (2023)). One kind of the hottest models is pre-trained transformer (Vaswani et al. (2017)) models, such as GPT-2 (Radford et al. (2019)), BERT (Devlin et al. (2018)), and VIT (Dosovitskiy et al. (2020)). After pre-training on related data, they perform well in generalization and could be quickly fine-tuned to downstream tasks.

When it comes to the defense for deployed service models, the condition would be harder. Facing an attack, the service model may be challenging to fine-tune since the methods, such as pruning (Zhu et al. (2021)), are usually implemented before deployment to compress or speed up the service. Thus,

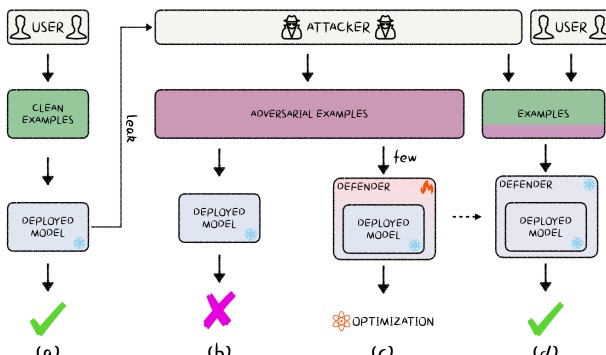

Figure 1: A case for rapidly defending a deployed model against the adversarial attack. (a) The deployed model is considered as a service for a certain task. (b) When some information about the service model is leaked, the attacker could generate adversarial examples by one attack method to fool the service model. (c) With a small number of adversarial examples, an adaptive defender is needed to avoid losses as quickly as possible. We suppose the following cases: the original deployed model is frozen since it is hard to tune and deploy quickly and well, little knowledge of clean data may be available, and few adversarial examples are available. (d) Equipped with the defender, the service could work correctly even with adversarial examples.

it costs a large computational budget to retrain and redeploy a more robust model. In addition, it's likely that only a small number of examples are possibly available. Moreover, we have to defend as quickly as possible to avoid more losses instead of waiting for getting enough training adversarial examples. Besides, clean data or the abstract knowledge of clean data, such as other models trained on it, is likely to be inaccessible. Therefore, recent works could not work in this case.

Under the mentioned difficulties and limitations, it is important to come up with a **Ra**pid **P**lug-**in** **D**efender (**RaPiD**). As shown in Figure 1, to simulate the conditions mentioned above, the victim service model is fixed, little knowledge of clean data and few possibly imbalanced adversarial examples of one attack method are available for training. In this paper, by default, only one-shot imbalanced adversarial examples are available unless stated otherwise. The main differences between **RaPiD** and the recent methods are shown in Table 1.

## 2    RELATED WORKS

**Adversarial Examples and Defenses.**    Introduced by Szegedy et al. (2013), adversarial examples could fool a neural network into working incorrectly. Among various methods (Akhtar & Mian (2018); Chakraborty et al. (2018)), attacks in a white-box manner are usually the most dangerous since the leaked information of the victim model is utilized. Many efforts generate adversarial examples through gradients of victims. Goodfellow et al. (2014) yielded a simple and fast method of generating adversarial examples (FGSM). Carlini & Wagner (2017) proposed much more effective attacks tailored to three distance metrics. PGD is a multi-step FGSM with the maximum distortion limitation (Madry et al. (2017)). Croce & Hein (2020) came up with AutoAttack, a parameter-free ensemble of attacks. Facing adversarial examples, lots of effort pay attention to defense. Some works strengthen robustness by adversarial training, where the model would be trained on adversarial examples (Goodfellow et al. (2014)). Wang et al. (2023) proposed to exploit diffusion models to generate much extra data for adversarial training. Xu et al. (2023) encouraged the decision boundary to engage in movement that prioritizes increasing smaller margins. In addition, many works focus on adversarial purification. Shi et al. (2021) combined canonical supervised learning with self-supervised representation learning to purify adversarial examples at test time. Similar to Wang et al. (2022), Nie et al. (2022) followed a forward diffusion process to add noise and recover the clean examples through a reverse generative process.

**Pre-trained Transformer.**    Introduced by Vaswani et al. (2017), transformer is an efficient network architecture based solely on attention mechanisms. It is first applied in natural language processing and then rapidly spread in computer vision. Devlin et al. (2018) proposed BERT to utilize only the

encoder of transformer while GPT-2 (Radford et al. (2019)) considered only transformer decoder. In computer vision, Dosovitskiy et al. (2020) proposed Vision Transformer (VIT), transforming a image into sequences of patches and processing them through a pure encoder-only transformer. Moreover, transformer has the ability of universal computation over single modality. Lu et al. (2021) demonstrated transformer models pre-trained on natural language could be transferred to tasks of other modalities. Similar to Zhu et al. (2023) and Ye et al. (2023), Tsimpoukelli et al. (2021) proposed to make the frozen language transformer perceive images by only training a vision encoder as the sequence embedding.

# 3 PRE-TRAINED TRANSFORMERS AS DEFENDERS

In **RaPiD**, with some adversarial examples, the defender should rapidly respond, keeping the original service fixed. We only consider image classification as the service task in this paper, but other tasks are also theoretically feasible. Motivated by the generalization and the universal computation ability of pre-trained transformer models (Lu et al. (2021); Kim et al. (2022)) and the case that pre-training could strengthen the robustness (Hendrycks et al. (2019)), we propose a new defender method, **CeTaD**, **C**onsidering **Pr**e-trained **T**ransformers **a**s **D**efenders, as shown in Figure 2. The plug-in defender is initialized by the pre-trained weights. A defender embedding and a defender decoder are needed to align the plug-in defender to the input example and the service model. There is a residual connection of the defender to

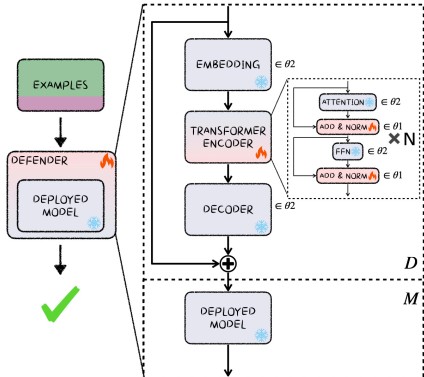

Figure 2: The structure of **CeTaD**. The input example would be added with the feature obtained by the stack of an embedding, a transformer encoder, and a decoder before being processed by the deployed service model. The deployed model is frozen in **RaPiD**.

keep the main features of the input example, which means that the original input example added with the output of the defender is the input for the service model. In this paper, the embedding is copied from VIT or BERT, and the decoder is PixelShuffle (Shi et al. (2016)). Since only limited adversarial examples are accessible, to avoid over-fitting and causing much bias on clean data, we choose to fine-tune minimal parameters, such as layer norm, of the plug-in defender. **CeTaD** is feasible for an arbitrary victim structure as long as it is differentiable.

Next, we formulate the method. In a single-label image classification task, every image $x_c$ among the clean set $\mathbf{X_c}$ is attached with a label $y^*$ among the corresponding label set $\mathbf{Y}^*$. A deployed model $\mathbf{M}$ maps $x_c$ into the prediction $y_c$ as

$$y_c = \mathbf{M}(x_c)$$

If $\mathbf{M}$ works correctly, $y_c = y^*$. Based on the leaked information of $\mathbf{M}$, the attacker edits the original image $x_c$ to an adversarial image $x_a$ by adding noises. $x_a$ belongs to the adversarial set $X_a$. The prediction for $x_a$ is

$$y_a = \mathbf{M}(x_a)$$

If the attack succeeds, $y_a \neq y^*$. The tuning set for defense is $\mathbf{X_d}$, which is the subset of $\mathbf{X_a}$. $|\mathbf{X_d}|$ is limited since adversarial examples are difficult to get.

In our method, we add a defender module $\mathbf{D}$ with parameters $\theta$ and keep $\mathbf{M}$ fixed. As shown in Figure 2, $\mathbf{M}$ consists of the embedding of a pre-trained VIT, a pre-trained transformer encoder as a feature poccessor and a parameter-free PixelShuffle block as a decoder. Only limited parameters are fine-tuned in $\mathbf{X_d}$. The objective is

$$\arg\min_{\theta_1} \sum_{x_d \in \mathbf{X_d}} loss(\mathbf{M}(\mathbf{D}_{\theta_1,\theta_2}(x_d) + x_d), y^*)$$

where *loss* is the cross-entropy for classification. $\theta_1$ and $\theta_2$ are the parameters of $\mathbf{D}$. Only $\theta_1$ is tuned. Specifically, layer norm parameters are $\theta_1$ and the others are $\theta_2$. With the trained defender $\mathbf{D}_{\theta_1^*,\theta_2}$, the final prediction $y$ is

$$y = \mathbf{M}(\mathbf{D}_{\theta_1^*,\theta_2}(x') + x')$$

where $\theta_1^*$ is the optimized parameters, and $x' \in (\mathbf{X_c} \bigcup \mathbf{X_a})$.

Here are two points of view on **CeTaD**. First, it could be considered a purifier, which perceives and filters the perturbations of adversarial examples by adding adaptive noise. From another angle, similar to prompt engineering (Liu et al. (2023)) in natural language processing, if we consider **CeTaD** as a prompt generator, it would generate adaptive prompts. The added prompts hint at the service model to better classify the adversarial examples.

## 4    EXPERIMENTS

### 4.1    EXPERIMENTAL SETUP

**Datasets and Attacks**    Three common datasets on image classification are considered: MNIST (LeCun et al. (2010)), CIFAR-10 (Krizhevsky (2009)), and CIFAR-100 (Krizhevsky (2009)). Two evasion-based methods, PGD (Madry et al. (2017)) and AutoAttack (Croce & Hein (2020)), are implemented to simulate attacks when a service model is leaked. Following Wang et al. (2023), maximum distortion $\epsilon$ is 8/255 for $l_\infty$-norm and 128/255 for $l_2$-norm. For PGD, the number of iterations is ten while the attack step size is $\epsilon/4$.

**Pre-trained Models**    For reproducibility, models and pre-trained checkpoints in the experiments are all public on GitHub or Huggingface. For MNIST, the victim model is a fine-tuned VIT-base; for CIFAR-10, both of a fine-tuned VIT-base and a standardly trained WideResNet-28-10 are considered as victims; For CIFAR-100, a fine-tuned VIT-base is the victim. Pre-trained BERT-base, BERT-large, VIT-base, VIT-large and GPT-2-124M are considered as the choices of the defender initialization. Here, we consider GPT-2-124M as a transformer encoder in **CeTaD** since it is to perceive information and following it, a decoder is implemented for mapping hidden feature into image space.

**Experimental Details**    In experiments, for simplicity, the training set only consists of adversarial examples whose number equals to that of the classes, namely one-shot; following Nie et al. (2022), we evaluate the accuracy on a fixed subset of 512 images randomly sampled from whole test data; by default, BERT-base is the defender for the WideResNet-28-10 against Linf-PGD on CIFAR-10; the embedding of the defender is taken from the pre-trained VIT; similar to Xie et al. (2022), the decoder is just implemented by PixelShuffle (Shi et al. (2016)) for less tuned parameters; only layer norm parameters of the defender is tuned while other parameters are completely frozen; Cross-entropy loss and Lion (Chen et al. (2023)) with default hyper-parameters is implemented for optimization; epoch is 500 and batch size is 32; Clean accuracy (CA), which stands for the accuracy on clean data without attack, and adversarial accuracy (AA), which stands for the accuracy on data with adversarial perturbations added, are considered to evaluate the defenders; following Lu et al. (2021), due to the number of experiments, we use one seed (42) for each reported accuracy in the content; unless stated otherwise. Each experiment could run on one NVIDIA RTX A5000 GPU within half an hour.

### 4.2    CAN PRE-TRAINED MODELS BE CONSIDERED AS DEFENDERS?

We investigate if a model pre-trained on another task could be considered as a defender. To do this, we apply **CeTaD** to MNIST, CIFAR-10, and CIFAR-100 datasets with the default settings mentioned in Section 4.1.

As shown in Table 2, without a defender, the original service model completely breaks down after performing attacks. Instead, although only limited parameters could be tuned and only one-shot

adversarial examples are available, models with **CeTaD** could correctly classify some adversarial examples. **CeTaD** is able to defend for both of VIT and ResNet on CIFAR-10, which shows that it is feasible for different victims. Besides, Both BERT and VIT defenders work, which may demonstrate that the frozen modules trained on the arbitrary dataset can be universal computation blocks and be aligned to defense, similar to Lu et al. (2021) and Kim et al. (2022).

In general, the performance for defense depends on the dataset and the defender initialization. Specifically speaking, for MNIST, the pixels of a number are relatively clear, and the background is always monotonous, which makes it easy to perceive the feature of adversarial perturbations. Thus, both of the defenders work well. However, when it comes to CIFAR-10 and CIFAR-100, the scene is more varied and complex. Tuning creates more bias, leading to the loss of clean accuracy. It is remarkable that VIT defenders outperform BERT defenders on clean accuracy while BERT defenders usually outperform VIT defenders on adversarial accuracy. The reason is that, for defense on image classification, the parameters of pre-trained VIT

Table 2: Accuracy performance of our method on different datasets. *None* represents no defense strategy.

| Dataset | Model | Defender | CA(%) | AA(%) |
|---------|-------|----------|-------|-------|
| MNIST | VIT | *None* | 98.83 | 00.78 |
| | | BERT | 98.05 | 92.77 |
| | | VIT | 98.24 | 91.41 |
| CIFAR-10 | ResNet | *None* | 93.75 | 00.00 |
| | | BERT | 68.75 | 44.34 |
| | | VIT | 82.81 | 30.27 |
| | VIT | *None* | 98.05 | 00.00 |
| | | BERT | 41.80 | 36.33 |
| | | VIT | 80.86 | 45.90 |
| CIFAR-100 | VIT | *None* | 91.41 | 00.00 |
| | | BERT | 44.53 | 34.77 |
| | | VIT | 52.34 | 30.47 |

are more stable since the original training task in VIT is similar to our test case, making it more vulnerable to adversarial perturbations. In contrast, the parameters of pre-trained BERT are more robust since the original training task is entirely different, making it challenging to classify clean examples.

Even if clean examples and adversarial examples are similar for humans, there is a wide gap for network models. Since only one-shot adversarial examples are available, the performance on clean data drops because of catastrophic forgetting (Goodfellow et al. (2013)). From another angle, considering the defender as a prompt generator, the prompts added into examples hint that the service model pays attention to adversarial features, leading to ignoring some clean features.

### 4.3 HOW IMPORTANT ARE THE DEFENDER STRUCTURES?

Though we find that **CeTaD** could work on different datasets, is the structure redundant? Here, we compare **CeTaD** with other possible structures and feasible state-of-the-art baselines for **RaPiD**. The methods are divided into 2 categories. Ones (R&P and Random Noise) are training-free while the others (Linear, FFN, Bottleneck and FD) need optimization. For R&P (Xie et al. (2017)), random resizing and random padding are utilized to defend against adversarial examples. For Random Noise, noise sampled from a normal distribution with a mean of zero is added to each test example as a defense, which is similar to BaRT (Qin et al. (2021)). FD (Xie et al. (2019)) utilizes a non-local denoising operation with a 1×1 convolution and an identity skip connection. We get both the best clean and adversarial accuracy for FD when the hidden dimension is set to

Table 3: Accuracy performance with different defense methods. Random Noise is similar to BaRT (Qin et al. (2021)).

| Method | CA(%) | AA(%) |
|--------|-------|-------|
| *None* | 93.75 | 00.00 |
| R&P (Xie et al. (2017)) | 93.16 | 02.34 |
| Random Noise(std=0.05) | 68.95 | 05.86 |
| Random Noise(std=0.06) | 57.23 | 11.13 |
| Random Noise(std=0.07) | 48.24 | 13.67 |
| Linear | 23.44 | 21.68 |
| FFN | 18.95 | 19.34 |
| Bottleneck | 23.44 | 20.90 |
| FD(Xie et al. (2019)) | 37.50 | 23.83 |
| GPT-2 (ours) | 55.08 | 39.65 |
| VIT (ours) | 82.81 | 30.27 |
| VIT-large (ours) | 71.68 | 44.14 |
| BERT (ours) | 68.75 | 44.34 |
| BERT-large (ours) | 66.02 | 48.83 |

256. For the Linear case, one linear layer without an activation function replaces the transformer layers. Similarly, FFN means one feed-forward block consists of two linear layers, the hidden feature dimension of which is double the input feature dimension, and one RELU activation function between them. The only difference between Bottleneck and FFN is that the hidden feature dimension is half of the input feature dimension for Bottleneck. It is worth mentioning that many previous methods on adversarial training, such as Wang et al. (2023), can not rapidly defense with limited adversarial examples due to their needs of abundant adversarial training data and serious time consumption for retraining the deployed model. Thus, they are not comparable with our method. In addition, since the

clean accuracy is rather high at the beginning of the training with random initialization, zero output initialization (Hu et al. (2021); Zhang & Agrawala (2023)) is not implemented.

The results are shown in Table 3. R&P maintains the clean accuracy but has little effect on the adversarial accuracy improvement. For the case of adding random noise, adversarial accuracy slightly increases, but clean accuracy seriously drops. In general, regarding adversarial accuracy, training-free methods are worse than those with optimization. Linear, FFN, and Bottleneck perform similarly. Because of the fixed effective denoising structure and the limited tuned parameters, FD is the best among the shown previous methods. However, compared with the methods above, **CeTaD**, initialized by GPT-2, VIT, VIT-large, BERT, or BERT-large, outperforms on adversarial accuracy while keeps rather high clean

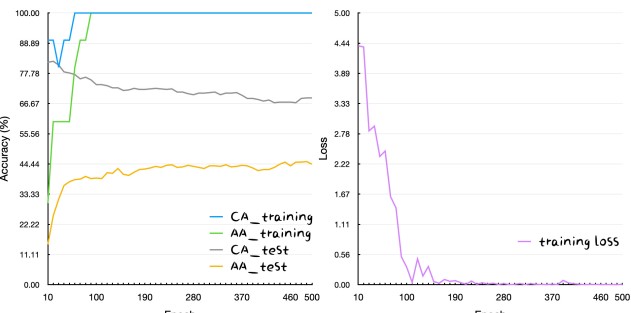

Figure 3: Accuracy and loss vs. epoch. **Left**: Accuracy curves on training and test data. *training* means it is on training data while *test* means on test data. It is worth mentioning that clean training data is actually unseen when training. **Right**: The loss curve on training data. Since the accuracy of training data is always 100% when the number of epochs is over 90, this loss curve is to better understand the training process.

accuracy. In addition, the defender initialized from GPT-2 is relatively poor. It demonstrates that although the decoder-based GPT-2 is efficient for many text tasks, combining the information for both former and latter patches in vision might be needed. It is also evident that the scale matters. The defenders of the large scale are better than those of the corresponding base scale in terms of adversarial accuracy.

In addition, when designing a defender, minimal tuned parameters and robustness of it are very essential. Linear, FFN, and Bottleneck are more flexible with much more tuned parameters when training, causing a trend to bias on the clean data. For **CeTaD**, since the fixed blocks are trained on other tasks, they are more robust. In addition, fewer tuned parameters result in better clean accuracy. More explorations about the tuned parameters of **CeTaD** are in Section 4.5.

We also evaluate the function of the residual connection of **CeTaD**. In Table 4, without this module, both clean and adversarial accuracy nearly crash into random selection. It seems that, with few tuned parameters and only one-shot adversarial examples, the residual connection is significant for both clean and adversarial accuracy.

## 4.4 HOW IS THE TRAINING PROCESS GOING?

With most parameters frozen and little tuned, could **CeTaD** well fit adversarial examples? In addition, since the training data consists of only one-shot adversarial examples by default, could **CeTaD** get overfitting? To evaluate these questions, we record clean and adversarial accuracy on both training and test data following default experimental settings. However, the accuracy of training data is not likely to be expressive because of its limited quantity. To better observe the training process, we also record the training loss on training data.

As shown in Figure 3, first, adversarial accuracy on training data increases up to 100% within 90 epochs, which means **CeTaD** is able to quickly fit training data with only layer norm parameters being tuned. To our surprise, clean accuracy also concomitantly rises to 100%. It is because even if clean examples are not directly shown for our model, training on adversarial examples could dig out some features that could reflect the corresponding clean examples.

Besides, on test data, adversarial accuracy steadily grows, which demonstrates that **CeTaD** could generalize the information learned from only one-shot adversarial examples. At the same time, clean accuracy drops. The distributions and mapping relationship to task space between clean data and adversarial data are not completely overlapped because of the function of added adversarial perturbations. Thus, when training, for **CeTaD**, drawing closer to the adversarial data domain would distance from the clean one, resulting in a loss of accuracy on clean data.

In addition, for the last 400 epochs, as accuracy on training data keeping 100%, adversarial accuracy on test data continues slightly rising, the corresponding clean accuracy gently declining and the loss occasionally shaking. It means that, instead of overfitting, **CeTaD** keeps exploring and learning information about adversarial examples. It is indeed vital since, in **RaPiD**, with limited training data, the difficulty is avoiding overfitting when training because methods such as evaluation and early stopping are likely not available for restricted examples.

Table 4: Accuracy performance on the residual connection. *without-res* is for removing the residual connection.

| Defender | CA(%) | AA(%) |
|---|---|---|
| *None* | 93.75 | 00.00 |
| BERT | 68.75 | 44.34 |
| BERT-*without-res* | 11.13 | 10.55 |
| VIT | 82.81 | 30.27 |
| VIT-*without-res* | 12.89 | 12.89 |

Table 5: Accuracy performance with different initialization strategies and tuned parameters.

| Defender | CA(%) | AA(%) |
|---|---|---|
| *None* | 93.75 | 00.00 |
| Random | 52.93 | 42.39 |
| Random-Tune-All | 43.36 | 33.79 |
| BERT | 68.75 | 44.34 |
| BERT-Tune-All | 59.77 | 44.14 |
| VIT | 82.81 | 30.27 |
| VIT-Tune-All | 69.14 | 36.14 |

## 4.5 ARE PRE-TRAINED INITIALIZATION AND FROZEN PARAMETERS NECESSARY?

Section 4.3 shows that initialization strategies and tuned parameters are vital for defenders. Here, we investigate these factors inside **CeTaD**. The difference between the BERT defender and the VIT defender is the weight initialization, as the structures of transformer layers are the same.

As shown in Table 5, tuning all parameters would reduce both clean and adversarial accuracy, except for the VIT defender. In that case, since the fixed modules of pre-trained VIT are also about image classification, the mapping relationship of the defender with limited tuning is close to that of the victim service, which makes it also vulnerable to adversarial examples. Instead, Tuning all parameters of VIT could distance from the original mapping relationship strengthening robustness, resulting in the increase of adversarial accuracy. In addition, we find that the BERT defender performs the best on adversarial accuracy. The VIT defender is better on clean accuracy and even the defender with random initialization still outperforms the VIT defender on adversarial accuracy. Therefore, the defender with VIT initialization seems more likely to be suboptimized and conservative.

## 4.6 HOW DOES TRAINING DATA AFFECT PERFORMANCE?

By default, only one-shot adversarial examples are accessible, and the adversarial examples are not class-balanced. For example, only 10 adversarial examples sampled randomly are available on CIFAR-10. It is to simulate the conditions where a deployed service model is attacked and only limited adversarial examples are relabeled. To discover how the training dataset affects the performance of **CeTaD**, we relax the settings for evaluation.

As shown in Table 6, based on the default setting, either adding one-shot clean examples for auxiliary, considering four-shot adversarial examples, or just balancing the class of the training data could enhance both clean accuracy and adversarial accuracy. The conditions of establishing class-balanced data and adding clean examples to training data are more important for improving clean accuracy.

## 4.7 COULD THE PROPOSED DEFENDERS ALSO RESPOND TO DIFFERENT ATTACKS?

In reality, the deployed service model may be attacked by various methods. To determine whether the defenders are reliable, we apply different attack methods and maximum distortion types to evaluate

Table 6: Accuracy performance on different training data settings. *1adv* (*1clean*) means one-shot adversarial or clean examples. *Balanced* means the examples are class-balanced.

| Training Data | CA(%) | AA(%) |
|---|---|---|
| *1adv* | 68.75 | 44.34 |
| *1adv-1clean* | 76.76 | 48.24 |
| *4adv* | 70.12 | 50.20 |
| *1adv-Balanced* | 77.34 | 49.02 |

Table 7: Accuracy performance against different attack methods. *None* represents no attack method is applied.

| Attack Method | CA(%) | AA(%) |
|---|---|---|
| *None* | 93.75 | - |
| $l_\infty$-PGD | 68.75 | 44.34 |
| $l_\infty$-AutoAttack | 70.70 | 49.41 |
| $l_2$-PGD | 76.17 | 57.03 |
| $l_2$-AutoAttack | 73.44 | 61.33 |

the defenders under the default experimental settings. Table 7 demonstrates that **CeTaD** is adaptable, and it is noteworthy that they get better adversarial accuracy against AutoAttack. We find that, in AutoAttack, only Auto-PGD works since the included methods are applied in turn for ensemble and the victim always completely fail against just the first method, Auto-PGD, which is able to automatically adjust the step size to get the minimal efficient perturbations. However, seeking for the minimal perturbations might cause poor robustness of the perturbations themselves, which makes it easier to successfully defend against. Thus, to generate better perturbations, the balance of maximum distortion and perturbation effect is much important.

### 4.8 COULD THE DEFENDERS GENERALIZE TO DIFFERENT DATASETS WITHOUT RE-TUNING?

Table 8: Accuracy performance on zero-shot transfer from top to bottom. Source is the environment where the defender is tuned while target is the environment which the defender transfers to. *None* represents the defender is directly trained in the target environment without transfer.

| Target Data (Target Model) | Defender | Source Data (Source Model) | CA(%) | AA(%) |
|---|---|---|---|---|
| CIFAR-10 (ResNet) | BERT | *None* | 68.75 | 44.34 |
| | | CIFAR-100 (VIT) | 63.87 | 7.42 |
| | VIT | *None* | 82.81 | 30.27 |
| | | CIFAR-100 (VIT) | 69.73 | 7.42 |
| CIFAR-10 (VIT) | BERT | *None* | 41.80 | 36.33 |
| | | CIFAR-100 (VIT) | 73.63 | 51.17 |
| | VIT | *None* | 80.86 | 45.90 |
| | | CIFAR-100 (VIT) | 79.88 | 47.66 |
| MNIST (VIT) | BERT | *None* | 98.05 | 92.77 |
| | | CIFAR-10 (VIT) | 96.29 | 90.43 |
| | | CIFAR-100 (VIT) | 97.85 | 89.84 |
| | VIT | *None* | 98.24 | 91.41 |
| | | CIFAR-10 (VIT) | 97.66 | 87.50 |
| | | CIFAR-100 (VIT) | 97.66 | 86.91 |

Since pre-trained models are good at generalization (Kim et al. (2022); Hendrycks et al. (2019); Lu et al. (2021)), the tuned defenders are likely to have the potential for transfer. Here, we evaluate **CeTaD** on different transfer tasks without re-tuning. As shown in Table 8, considering ResNet on CIFAR-10 as the target and VIT on CIFAR-100 as the source, adversarial accuracy is even lower than that of random selection. If the target model is changed to VIT, **CeTaD** has better performance for transfer. Thus, since **CeTaD** is tuned end-to-end, they are sensitive to the structure of the victim service model and cannot directly transfer across

Table 9: Accuracy performance on zero-shot transfer from bottom to top.

| Defender | Source Data (Source Model) | CA(%) | AA(%) |
|---|---|---|---|
| BERT | *None* | 44.53 | 34.77 |
| | CIFAR-10 (VIT) | 13.87 | 12.89 |
| | MNIST (VIT) | 26.37 | 23.44 |
| VIT | *None* | 52.34 | 30.47 |
| | CIFAR-10 (VIT) | 45.31 | 27.54 |
| | MNIST VIT) | 49.41 | 28.91 |

different victim models. Instead, when the designs of the victim models are similar, the transfer from the source task to the target task may be beneficial. Specifically, The transferred BERT defender get higher adversarial accuracy than others. Thus, **CeTaD** tuned on much more complex data could perform better. In addition, since CIFAR-10 and CIFAR-100 are similar, we consider MNIST as the target data and CIFAR-10 or CIFAR-100 as the source data. The performance is comparable to that of direct tuning, and it is similar no matter whether the source data is CIFAR-10 or CIFAR-100, which means the knowledge of these defenders that could be transferred is identical.

The evaluations above are about transferring from a more challenging source data. It is much more meaningful when the target task is more challenging than the source tasks. As shown in Table 9,

CIFAR-100 is the target data since it is more complex. Surprisingly, the defenders tuned on MNIST have better adversarial accuracy than CIFAR-10. It illustrates that the transfer from unrelated data may be better than that from related data. The reason is that transfer from different domains would enhance the robustness of the defenders. To sum up, the transfer gap would improve the robustness of defense, so the defender on diverse datasets may further strengthen the ability on a single dataset.

## 5 DISCUSSION: LIMITATIONS AND FUTURE WORK

For now, in **RaPiD**, even if more powerful attacks have not been considered, there still is a significant distance from reliability for the performance of **CeTaD**. Tuning **CeTaD** end-to-end would more or less damage the performance on clean data. Since the clean and adversarial data are usually similar in pixels, maybe we could remain clean accuracy by digging out the feature of clean data left on adversarial data.

Though we only consider image classification in this paper, **CeTaD** is able to be applied into other differentiable systems. We are looking forward to evaluating the performance and the generalization in various tasks in future work. Moreover, is it possible to include methods, such as genetic algorithm and reinforcement learning, to break the limitation of differentiability?

In addition, as demonstrated in Section 4.5, the initialization strategy and tuned parameter selection would influence a lot. This paper evaluates only three initialization strategies from standard pre-trained models while only the case of tuning the parameters of layer norm and fine-tuning all defender parameters are considered. Therefore, a better initialization strategy for defense and the data-driven elaborate selection for tuned parameters could improve the performance.

Besides, the conditions of training data are also a vital factor. In this paper, most experiments consider only one-shot imbalanced adversarial examples as training data. However, as shown in Section 4.6, the class balance of adversarial examples and the mixture of adversarial examples and clean data could help a lot. Several adversarial examples and clean examples may be available. Thus, we may slightly relax the limitations in **RaPiD**, focusing on structuring a training set consisting of few-shot clean and adversarial examples with the minimal quantity to get the maximal performance.

Furthermore, lifelong learning should be considered. Though we only include one attack method in each experiment, a service model is like to be attacked by different methods from time to time. Thus, we need a defender which can continuously learn to defend against a new attack method while keep and even study from the learned knowledge in the past. By the way, we believe that the defense for deployed models is a complex system. Though we focus on the core (how to defend), there are many other unresolved important problems, such as how to rapidly detect adversarial examples when the attack happens.

In Section 4.8, it is surprising that indirectly related data transfer outperforms related data transfer even from bottom to top. This means there is consistency in different data though the specific domains differ. Thus, whether we could align modalities through such consistency is a good question. Furthermore, what about the transferability across different attack methods and how to well transfer across various victim models are left for future work. By including diverse service models for various tasks on multi-modality data against different attack methods, we are possibly able to get a relative universal defender, which could strengthen its robustness in one domain from others.

## 6 CONCLUSION

In this paper, we analyse the defense for deployed service model and find that the solutions in the case, **RaPiD**, are essential and related works can not work well. Leveraging the generalization and universal ability of pre-trained transformers, we propose **CeTaD**, a new defender method considering pre-trained transformers as defenders. In experiments, we demonstrate that our method could work for different victim model designs on different datasets against different attacks and explore the optimization process, initialization strategies, frozen parameters, structures, training data conditions and zero-shot generalization in our method.

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
