## A   SOCIAL IMPACTS

The tidal wave of applying and deploying deep neural networks is coming. We believe the studies on the safety of deep neural networks are becoming more and more vital and urgent. In this paper, we focus on an important case, **RaPiD**, where the previous methods are not able to provide rather effective defense, and explore a new defense framework, **CeTaD**. However, for now, our method is still not reliable enough and many limitations exist. Besides, the condition is more complex in practical terms. To avoid heavy losses, we recommend that, in industrial systems, before being applied, the defense methods should be carefully investigated and evaluated. We also encourage related researchers in the community to focus on more reliable and practical defense methods in the future.

## B   DETAILS OF DATA PREPARATIONS

For reproducibility, we illustrate how to prepare data in the experiments.

All datasets are available from Huggingface: MNIST (`https://huggingface.co/datasets/mnist`), CIFAR-10 (`https://huggingface.co/datasets/cifar10`) and CIFAR-100 (`https://huggingface.co/datasets/cifar100`). The library, *Datasets* (`https://github.com/huggingface/datasets`), which includes the methods mentioned below, is utilized for downloading and splitting data.

**N-shot Training Samples.**   First, we split data by class using *filter*. Then, for each category, two methods, *shuffle* with a given seed and *select* for getting the first $n$ samples, are applied in turn. Finally, we mix the selected samples of all classes by *concatenate_datasets* and *shuffle* with the seed.

**512 Fixed Test Samples.**   We apply *shuffle* with the seed and *select* to get the first 512 samples.

## C   DETAILS OF MODULE SELECTIONS INSIDE CETAD

Module selections are essential for **CeTaD** since only limited parameters are tuned. The embedding and the decoder are vital for feature mapping between the input space and the hidden space. The encoder is significant for perceiving adversarial information and enhancing robustness since it is the only trainable module and bears the most computation in **CeTaD**.

As shown in Figure 2 and illustrated in Section 3, **CeTaD** is flexible as long as the dimensions of the modules match with each other. However, pre-trained weights may help.

For example, we take the embedding from the pre-trained VIT, get the transformer blocks from the pre-trained BERT, VIT or GPT-2, and consider PixelShuffle as the decoder. The modules we used are briefly introduced as follows: BERT (Devlin et al. (2018)) is a transformer encoder model pre-trained for masked language modeling (MLM) and Next sentence prediction (NSP) on a large corpus of uncased English data (base: `https://huggingface.co/bert-base-uncased`; large: `https://huggingface.co/bert-large-uncased`); VIT (Dosovitskiy et al. (2020)) is a transformer encoder model pre-trained for image classification on ImageNet-21k at resolution 224x224 (base: `https://huggingface.co/google/vit-base-patch16-224-in21k`; large: `https://huggingface.co/google/vit-large-patch16-224-in21k`); GPT-2 (Radford et al. (2019)) is a transformer decoder model pre-trained for causal language modeling (CLM) on a large corpus of English data (124M: `https://huggingface.co/gpt2`); PixelShuffle (Shi et al. (2016)) rearranges elements unfolding channels to increase spatial resolution [1].

---

[1]In the experiments, *upscale_factor* is always set to 16. Thus, if the scale of the transformer encoder is large, which means the hidden feature is of 1024 dimensions and four channels are given after PixelShuffle, we just ignore the last channel for simplicity.

## D  DETAILS OF OPTIMIZATION

Optimization loops are implemented by PyTorch. To optimize limited parameters and freeze the others, following Lu et al. (2021), we set *requires_grad=True* for tunable parameters while *requires_grad=False* for the others. The optimizer is initialized by registering the parameters with *requires_grad=True*. Under the default experimental setup, only layer norm parameters (48 parameter groups, 36864 variables in total) are tuned.

By the way, the implementation of Lion (Chen et al. (2023)), the optimizer which we apply, is available at `https://github.com/lucidrains/lion-pytorch`.

## E  ERROR BARS

Following Nie et al. (2022), we evaluate the accuracy on a fixed subset of 512 images randomly sampled from whole test data to save computational cost. Besides, because of the number of experiments and the page limit, following Lu et al. (2021), in the content, we only report the results with one seed (42—*the answer to the ultimate question of life, the universe and everything*). In this section, to show the validity of the results in the content, we additionally repeat two experiments described in Section 4.2 and Section 4.4 with another two seeds (41 and 43).

In Table 2, Table 10 and Table 11, with a different seed, though the training data and the fixed subset for evaluation vary, leading to accuracy fluctuation, the relative performances of different methods remain the same. Specifically, as illustrated in Section 4.2, VIT defenders are better at clean accuracy while BERT defenders are likely to outperform at adversarial accuracy. Furthermore, the trends of the corresponding curves in Figure 3, Figure 4 and Figure 5 are similar. It demonstrates that our experiments are both efficient and effective.

Table 10: Accuracy performance with seed 41.

| Dataset | Model | Defender | CA(%) | AA(%) |
|---------|-------|----------|-------|-------|
| MNIST | VIT | *None* | 99.02 | 00.59 |
| | | BERT | 97.07 | 90.82 |
| | | VIT | 99.02 | 91.60 |
| CIFAR-10 | ResNet | *None* | 93.95 | 00.00 |
| | | BERT | 70.12 | 43.55 |
| | | VIT | 76.95 | 28.91 |
| | VIT | *None* | 97.85 | 00.00 |
| | | BERT | 35.94 | 31.84 |
| | | VIT | 76.37 | 41.60 |
| CIFAR-100 | VIT | *None* | 91.80 | 00.39 |
| | | BERT | 50.78 | 38.28 |
| | | VIT | 54.30 | 31.45 |

Table 11: Accuracy performance with seed 43.

| Dataset | Model | Defender | CA(%) | AA(%) |
|---------|-------|----------|-------|-------|
| MNIST | VIT | *None* | 99.22 | 00.59 |
| | | BERT | 98.83 | 93.36 |
| | | VIT | 99.22 | 87.70 |
| CIFAR-10 | ResNet | *None* | 95.51 | 00.00 |
| | | BERT | 73.05 | 44.73 |
| | | VIT | 79.30 | 32.81 |
| | VIT | *None* | 98.05 | 00.00 |
| | | BERT | 69.73 | 53.52 |
| | | VIT | 80.86 | 53.13 |
| CIFAR-100 | VIT | *None* | 94.14 | 00.20 |
| | | BERT | 44.14 | 34.18 |
| | | VIT | 47.07 | 28.32 |

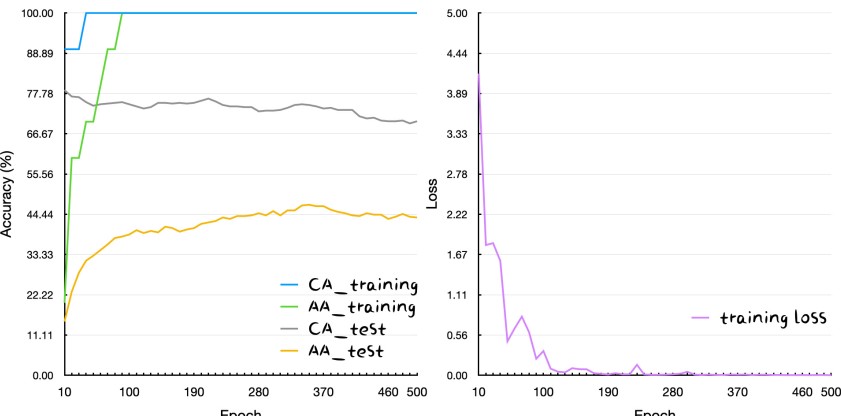

Figure 4: Accuracy and loss vs. epoch with seed 41.

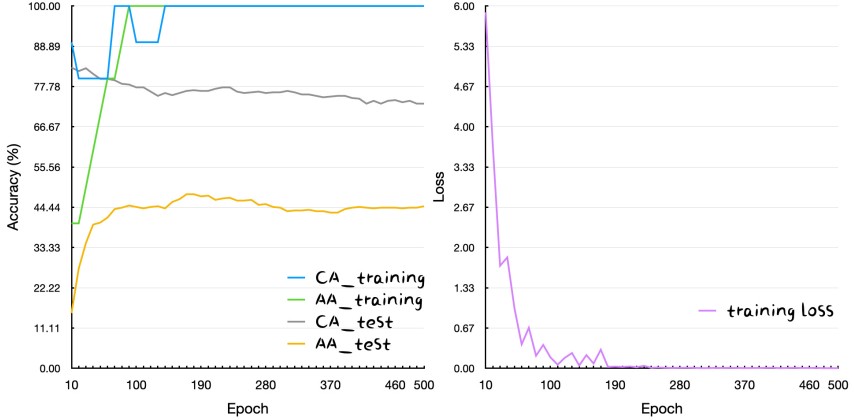

Figure 5: Accuracy and loss vs. epoch with seed 43.