# OpenReview forum: "Pre-trained Transformers as Plug-in Defenders Against Adversarial Perturbations"
_ICLR.cc/2024/Conference — ICLR 2024 Conference Withdrawn Submission_

### Official Review · Reviewer_GjLF · 2023-10-29

**Soundness:** 3 good
**Presentation:** 2 fair
**Contribution:** 2 fair
**Rating:** 3
**Confidence:** 2

**Summary:**

This paper proposes using pretrained transformers as a "purification" module to protect a fixed model against adversarial perturbations. The pretrained module will be fine-tuned using an objective described at the top of page 4, where the objective tries to patch a input so that a model gets correct prediction (y^*).

Some experiments have been performed to demonstrate the effectiveness of this method, with current limitations discussed towards the end of the manuscript.

**Strengths:**

Using pretrained-transformers as a purification layer seems new. The fine-tune objective is natural (on page 4), but also seems to be new. Quite some experiments have been performed on the effectiveness of the method.

**Weaknesses:**

My first objection comes from that the paper seems quite dismissive (no offense intended) about the "security model". The paper spends a lot of time discussing ML concepts, but when getting to talk about adversarial attacks, it didn't describe in enough detail how the attacks are performed.  For example, the only sentence I could really find that is explicitly about attacks is:

"""
three common datasets on image classification are considered: MNIST
(LeCun et al. (2010)), CIFAR-10 (Krizhevsky (2009)), and CIFAR-100 (Krizhevsky (2009)). Two
evasion-based methods, PGD (Madry et al. (2017)) and AutoAttack (Croce & Hein (2020)), are
implemented to simulate attacks when a service model is leaked
"""

So is the attacker only able to attack the underlying service model? Or that he can also have knowledge about the pretrained transformer? Is it completely white box (knowing everything including the defense mechanism), or black box (knows nothing except that the adversary can access the model through APIs, or somewhere in between?). Without knowing these it is really hard to evaluate the merits of the paper.

If we assume that the defender does not have knowledge about the defense/transformer, then it seems that we went back to the old issue about the purification approach: That if the adversary has knowledge about the purifcation would he easily break the defense. It might be useful to argue that the cost on working with a pretrained transformer is too high. But such things need to be discussed in the paper.

Aside from this, the authors also admitted in the limitations the current method seems to have a poor reliability on performance.

Overall, I found this paper quite premature for publication.

**Questions:**

No specific question. My main concerns have been listed above.

---

### Official Review · Reviewer_Lpvc · 2023-10-31

**Soundness:** 2 fair
**Presentation:** 2 fair
**Contribution:** 2 fair
**Rating:** 3
**Confidence:** 3

**Summary:**

This paper addresses the critical issue of the susceptibility of deep neural networks to adversarial attacks. Specifically, the paper examines evasion-based attacks that are particularly problematic for models deployed in real-world applications. While existing solutions often use adversarial training or rely on clean data to improve network robustness, these approaches are computationally expensive and not always feasible for deployed models. The authors propose an innovative solution that involves using pre-trained transformers as "plug-in defenders" to improve the resilience of neural networks.

**Strengths:**

The paper introduces a new approach for defense by using pre-trained transformers as plug-in defenders against adversarial attacks.

The paper considers a resource constrained setup which has some practical application where the defender does not have access to clean datasets.

The paper acknowledges its limitations and suggests directions for future research.

**Weaknesses:**

This paper only considers natural image classification tasks. How to adapt this approach for tasks such as Medical image classification tasks. The defender relies on VIT pretrained embedding. How would a defender get pre-trained embedding for Medical image classification tasks?

Imagenet dataset results are not reported. How does the method perform with the imagenet dataset?

It is not clear why BERT is able to filter out the adversarial noise from the adversarial inputs since BERT is pre-trained for NLP tasks. While the authors show that it works experimentally but the rationale behind this is missing.

Does fine-tuning the base model with the adversarial examples work without CeTaD. This performance must be reported since if this leads to on par performance then why would someone add CeTaD instead of fine tuning the base model.

This approach is going to increase the inference time of the service overall. The inference time with and without CeTaD must be reported.

**Questions:**

Please refer to the weakness section.

---

### Official Review · Reviewer_7fic · 2023-11-01

**Soundness:** 3 good
**Presentation:** 3 good
**Contribution:** 2 fair
**Rating:** 3
**Confidence:** 5

**Summary:**

This work proposed CeTaD, a new defender method considering pre-trained transformers as defenders. They did it by using a Encoder->Decoder (can be seen as a denoiser) before the frozen model. Parts of the denoiser are also frozen.

**Strengths:**

The paper shows you can get a fair amount of robustness with minimal training, while relaying on frozen layers.

**Weaknesses:**

My main concern for this paper is that there are many works in this field, and the authors did not mentioned or discussed none of them.
The idea of adding denoising layers before the model is not new, and authors should talk about the related work + compare to it.

Also, in (CERTIFIED!!) ADVERSARIAL ROBUSTNESS FOR FREE! (https://arxiv.org/pdf/2206.10550.pdf) Carlini et. al. Also experiment with the same idea of using pre-trained components (pretrained denoising diffusion probabilistic model and a standard high-accuracy classifier.) which is conceptually very similar to the idea proposed in this paper.

I suggest the authors to improve the related work section and show the results compared to other works that try to "plug-and-play" and works that use denoising components before the model.

**Questions:**

See Weaknesses section.

---

### Official Review · Reviewer_9SPS · 2023-11-02

**Soundness:** 3 good
**Presentation:** 3 good
**Contribution:** 1 poor
**Rating:** 3
**Confidence:** 4

**Summary:**

The paper proposes a parameter-efficient adversarial training approach for transformer based vision models. They show that the approach can boost the performance on adversarial attacks with some impact to classification accuracy.

**Strengths:**

- The paper is well-written, easy to understand and tests the major claims presented.
- The authors do a good job on data ablations and analyzing the effect of training data on model performance on both classification and adversarial accuracy.

**Weaknesses:**

1. The model design resembles that of an adapter [1,2]. In adapters, a set of weights are added to different layers of a transformer and only these additional are weights are fine-tunes on a smaller dataset. How is this work different from considering the concept of adapters for adversarial examples? If it is not, the authors should portray it at such as opposed to brining in new terminology such as RaPiD and CeTaD (which can be called as parameter efficient fine-tuning with adapters for adversarial examples). Even if the proposed parameters design is different from adapters, a comparison to the baseline would be necessary, esp given the next point.
2. There is already existing works on learning adapters on adversarial examples in Natural Language [3,4]. At the very least, the authors should compare and contrast the novelty of their approach w.r.t. these given they themselves claim their approach is general and can be considered for non-vision tasks (similarly, [3] and [4] can be considered for vision). This raises question on novelty and awareness of the authors about relevant related work.
3. The authors say "Besides, clean data or the abstract knowledge of clean data, such as other models trained on it, is likely to be inaccessible." Not sure why this is likely to be inaccessible to the defender? While it is reasonable to frame the treat-model for the proposed solution, the authors seem to critic existing threat-models unnecessarily without concrete evidence (eg. (1) sample/data complexity of their method vs adversarial training, (2) why not simply use the adversarial examples as part of training data, when detected, for continual fine-tuning, etc.)
4. Is there a reason to chose the one-shot per class setting? As in, why is it easier to get one adversarial example per class in the wild for motivating the parameter efficient setup? Would it be better to obtain multiple samples for sub-set of common classes?
5. The one-shot adversarial example used for parameter-efficient finetuning (called CeTaD), is it generated using PGD or AutoAttack or a mix of those? If generated using PGD, how is the out-of-distribution performance on AutoAttack examples at test time and vice versa?
6. The drop in classification accuracy using BERT and VIT as the adapters/defenders (ResNET/VIT accuracies on CIFAR-10 drops from 93.27 to 68/42 and 83/81 resp; CIFAR-100 drops from 91 to 44 and 52 resp.) seems to suggest that the adapter approach can improve on the adversarial attacks, but results in classifiers that are no more useful for the actual task (Table 2). This drastically weakens the proposed appraoch.
6. Did you happen to test the performance of these models on black-box attacks or universal perturbations?

> [1] Houlsby, N., Giurgiu, A., Jastrzebski, S., Morrone, B., De Laroussilhe, Q., Gesmundo, A., ... & Gelly, S. (2019, May). Parameter-efficient transfer learning for NLP. In International Conference on Machine Learning (pp. 2790-2799). PMLR.

> [2] Chen, Z., Duan, Y., Wang, W., He, J., Lu, T., Dai, J., & Qiao, Y. (2022, September). Vision Transformer Adapter for Dense Predictions. In The Eleventh International Conference on Learning Representations.

> [3] Rebuffi, S. A., Croce, F., & Gowal, S. (2022, September). Revisiting adapters with adversarial training. In The Eleventh International Conference on Learning Representations.

> [4] Han, W., Pang, B., & Wu, Y. N. (2021, August). Robust Transfer Learning with Pretrained Language Models through Adapters. In Proceedings of the 59th Annual Meeting of the Association for Computational Linguistics and the 11th International Joint Conference on Natural Language Processing (Volume 2: Short Papers) (pp. 854-861).

**Questions:**

See above.